# Differences in Children and Adolescents with Depression before and after a Remediation Program: An Event-Related Potential Study

**DOI:** 10.3390/brainsci14070660

**Published:** 2024-06-28

**Authors:** Nikolaos C. Zygouris

**Affiliations:** Digital Neuropsychological Assessment Laboratory, Department of Informatics and Telecommunications, University of Thessaly, 35100 Lamia, Greece; nzygouris@uth.gr; Tel.: +30-2231060184

**Keywords:** children and adolescent depression, cognitive behavioral therapy, event-related potentials, remediation depression, P300 latency

## Abstract

Depression is clinically diagnosed when a defined constellation of symptoms manifests over a specific duration with notable severity. According to the Diagnostic and Statistical Manual of Mental Disorders, Fifth Edition (DSM-5), Major Depressive Disorder (MDD) is characterized by the presence of five or more symptoms persisting for at least two weeks. As a profound mental health condition affecting millions globally, depression presents a considerable challenge for researchers and clinicians alike. In pediatric and adolescent populations, depression can precipitate adverse outcomes, including substance abuse, academic difficulties, risky sexual behaviors, physical health problems, impaired social relationships, and a markedly elevated risk of suicide—up to thirty times higher than the general population. This paper details a study that evaluated the efficacy of Cognitive Behavioral Therapy (CBT) alone vs. CBT combined with selective serotonin reuptake inhibitors (SSRIs) in a treatment program. The study cohort comprised sixteen (16) children and adolescents diagnosed with depression (eight males and eight females) and sixteen (16) typically developing peers (eight males and eight females) aged from 9 to 15 years (Mean age = 11.94, standard deviation = 2.02). Initial assessments employed Event-Related Potentials (ERPs), the Children’s Depression Inventory (CDI), and reaction time measurements. The results reveal that participants with depression exhibit cognitive deficits in attention and memory, as evidenced by prolonged P300 latencies. Following intervention with either CBT alone or CBT combined with medication, the depressed participants demonstrated significant improvements, evidenced by lower CDI scores, reduced P300 latencies, and faster reaction times, both compared to their pre-treatment status and relative to the control group.

## 1. Introduction

Major Depressive Disorder (MDD) is a prevalent mental disorder among children and adolescents. The three-month prevalence rates for MDD in children and adolescents aged 9–16 are 2.2% overall, with 2.8% for males and 1.6% for females. By age 16, the cumulative prevalence rate is projected to reach 9.5% overall, 11.7% in females, and 7.3% in males [1]. A meta-analysis has estimated that the prevalence of depression in children under 13 is 2.8%, while 5.6% of adolescents aged 13–18 experience depression, with a higher prevalence in females (5.9%) compared to males (4.6%) [2]. Additionally, another study has identified that the onset of mental health disorders, including depression, peaks around 14 years of age [3]. According to the World Health Organization, depression and anxiety disorders rank among the top five causes of psychopathologies in European children and adolescents [4]. It must be noted that in a meta-analysis [5], the prevalence of clinically elevated depression symptoms during the COVID-19 pandemic was 25.2%. Consequently, one in four young people worldwide is experiencing clinically elevated depression symptoms.

According to the DSM-5 [6], cognitive impairments are a primary symptom of depression, characterized by a reduction in the ability to think, focus, or make judgments. Cognitive performance in patients with MDD is crucial, as these impairments correlate with a diminished ability to function in daily life. Previous studies have identified deficits in executive function, memory, psychomotor speed, and attention in MDD patients compared to healthy controls [7]. These deficits can range from mild to severe and may persist even after the depressive episode has been treated [8]. Notably, impairments in attention, executive function, and memory have been observed even after symptom remission, suggesting that cognitive impairments may be trait markers rather than mere symptoms of depression [9]. Although research on the relationship between symptom severity and cognitive impairment has yielded ambiguous results, a positive correlation is suggested [10].

In vivo neuroimaging studies have revealed correlations between depression and changes in brain structure and function. Research has connected the amygdala-medial prefrontal cortex circuitry to emotional [11] and social processing [12], leading to the concept that dysfunctional interactions between cortical and subcortical systems cause depression [13]. Depression is associated with disrupted metabolism and altered gray matter volume in the prefrontal cortex, which may correlate with illness duration [14]. Furthermore, neuroimaging studies have shown minor patterns of cortical thinning in the medial prefrontal cortex, subgenual anterior cingulate cortex, and ventral temporal lobes, which correlate with illness severity [15]. Depression is related to reductions in global brain connectivity in these regions, extending to dispersed portions of the multimodal association cortex [16].

The salience network, which includes cortical and limbic brain areas such as the amygdala–hippocampal complex, is activated by emotionally salient stimuli and tends to perpetuate a negative effect in depressed individuals. The amygdala is hyperactive during negative stimuli but hypoactive during positive ones. Adolescents with severe depressive disorders have shown reduced activity in the amygdala–hippocampal complex during positive self-processing tasks compared to neutral-processing tasks [17]. The biological causes of these changes may be related to GABAergic changes [18] or decreased size and density of neurons and glia, particularly astrocytes [19]. Serotonergic dysfunction in the central nervous system is considered a significant pathophysiological factor in depression [20], prompting the investigation of ERPs in patients with MDD. However, it is clear that, besides serotonin, other neurotransmitters, such as noradrenaline and dopamine, also contribute to the development of depressive disorders [21]. Different subtypes of depression may vary in neurotransmitter function, suggesting that ERPs may serve as a preferred method for studying depressive disorders.

Event-Related Potentials (ERPs) are a non-invasive electrophysiological procedure that provides information about brain activity associated with cognitive information processing. ERPs have been widely used to study the correlations between the electrical activity of the brain and cognitive processes [22]. The P300 component of ERPs, in particular, is associated with higher-order cognitive information processing and attentional tasks linked with contextual evaluation [23]. P300 latency is suggested to reflect higher-order cognitive processes such as classification and stimulus appraisal, serving as a temporal signal of brain activity related to attention allocation and memory processes [24].

Studies using ERPs report that depressed individuals often present with lower P300 amplitudes and prolonged P300 latencies compared to healthy controls, suggesting significant differences in cognitive processing [25,26,27,28,29,30,31,32,33,34,35,36,37]. However, relevant research has found no significant difference in the latency of the P300 waveform [24]. These discrepancies may be due to methodological variations and the inclusion of diverse patient groups, including those receiving pharmacological and psychotherapeutic interventions. Nonetheless, numerous studies have reported alterations in P300 amplitude and latency in individuals with depression, indicating disruptions in cognitive processing and attentional mechanisms (for review [33]). 

Electrophysiological studies have, thus, provided valuable insights into the neural mechanisms underlying depression, revealing abnormalities in brain activity that contribute to the disorder’s cognitive, affective, and behavioral symptoms [35] (for review [38]). Leveraging neuroscience to explore cognition, behavior, and the impact of the environment greatly contributes to developing and accessing personalized treatments and clinical methodologies [34]. Brain plasticity, characterized by rapid and reversible alterations in brain function and structure, manifests in learning, memory, and perception. Changes in synaptic plasticity signify learning and memory on a cellular level, indicating that neurons can adjust the strength and arrangement of their synapses based on experience [36].

Cognitive Behavioral Therapy (CBT) is an empirically supported psychological treatment for a variety of disorders, including depression [37,39]. CBT involves cognitive reconstruction, behavioral change, and social support, aiming to help individuals identify stressors and modify negative cognitive beliefs and behaviors, thereby reducing or eliminating symptoms of psychological distress and resuming normal psychological and social functions [39]. Group and individual CBTs have similar content, with group CBT offering additional benefits such as peer modeling, practice of interpersonal and communication skills, feedback, positive reinforcement, and social comparisons [40]. Numerous studies have demonstrated the efficacy of both CBT methods in treating depressive disorders. A review and meta-analysis [38] found that CBT is an effective treatment for depression and that combining it with medication is more beneficial than pharmacotherapy alone. Additionally, evidence suggests that patients treated with CBT have a lower relapse rate compared to those treated with medication alone [41,42,43,44,45,46,47,48,49,50]. 

The study by Dickey et al. [51] combined CBT with electrophysiological results, suggesting that individual differences in neural and self-reported measures of reward responsiveness and emotion regulation can predict changes in depressive symptoms and clinician-rated improvement. The findings indicate that adolescents who had reduced reward-positive responses to reward feedback before treatment, indicating diminished neural responsiveness to rewards, reported fewer depressive symptoms after treatment, taking baseline symptoms into account. Additionally, adolescents with larger frontal late positive potential residuals during the reappraisal of dysphoric images, possibly indicating impaired emotion regulation, showed greater clinician-rated improvement with CBT among those who completed the treatment. Neural measures were more predictive of treatment outcomes than clinical and self-report predictors, highlighting their potential clinical utility. Their findings are supported by several studies [51,52,53,54,55,56] using both CBT and SSRIs in the treatment of patients with depression.

The primary objective of this research is to contrast the electrophysiological brain responses of children and adolescents diagnosed with depression for the first time. We rely on experiments performed at a state child and adolescent psychiatric hospital compared with typically developing peers using ERPs, specifically examining the P300 latency. Subsequently, the group of depressed children and adolescents participated in a remedial program involving either CBT alone or a combination of CBT and psychopharmacological support. Their brain activity was recorded upon the completion of the program. Both the depressed and control groups underwent behavioral assessments, including the Greek version of the Children’s Depression Inventory (CDI) [57] and reaction time measurements.

Consistent with previous studies [25,26,27,28,29,30,31,32,33,34,35,36,37], the first hypothesis was that children and adolescents with depression would exhibit longer P300 latencies, slower reaction times, and higher CDI scores compared to the control group. 

The second hypothesis was that following the remediation program, the depressed group would show decreased P300 latencies, faster reaction times, and lower CDI scores compared to pre-remediation measurements (for review [45,46]). The third hypothesis was that the type of remediation program (CBT alone or combined with psychopharmacological support) would result in different P300 latency and reaction time outcomes [51,52,53,54,55,56]. 

## 2. Materials and Methods

### 2.1. Participants

Sixteen (16) right-handed children and adolescents (8 males and 8 females) aged 9–15 years (M = 11.94, standard deviation—SD = 2.02) diagnosed with depression by the Athens Children and Adolescent Psychiatric Hospital participated in this study. None of the participants in the depression group had previously completed any therapeutic program, as they were receiving assessment and diagnosis for the first time. The control group comprised sixteen right-handed children and adolescents of the same age (M = 11.74, SD = 3.10) and gender (8 males and 8 females) as the depression participants, with no psychopathological difficulties. School medical records confirmed that none of the participants, whether in the control or depressed groups, had learning difficulties, developmental disorders, or significant visual or hearing impairments. Participants were recruited through informative newspaper articles, study notifications from the hospital, and informative school sessions. Depression diagnoses were made using standard assessment protocols, including a clinical interview with the Greek version of K-SADS-PL DSM-5 [58], conducted by a child and adolescent psychiatrist in accordance with DSM-5, and responses to the Greek version of the Children’s Depression Inventory (CDI) [57]. It must be highlighted that all children with depression scored highly in CDI questions that assess affective behavior, ideation (image), interpersonal relations, and guilt/irritability. All participants’ parents/guardians signed consent forms permitting their child’s participation. All human data included in this manuscript were obtained in compliance with the Helsinki Declaration and the guidelines of the Ethics and Deontology Committee of the University of Thessaly.

### 2.2. Electrophysiological Assessment

A passive auditory oddball paradigm was used to investigate P300. Participants sat comfortably in a reclining armchair with their eyes closed while receiving 200 auditory stimuli delivered binaurally through headphones. The stimuli consisted of brief (40 ms), 85 dB intensity tones with 10-ms rise and fall times. An interstimulus interval (ISI) of 1500 ms separated each tone presentation. Two types of stimuli were used: non-target stimuli with a frequency of 1000 Hz and target (oddball) stimuli with a frequency of 2000 Hz. Participants were instructed to use their right hand to press a button as quickly and accurately as possible upon detecting the target stimulus. Reaction times were recorded [59].

ERPs were measured using a Medtronic device (710 Medtronic Parkway, Minneapolis, MN, USA). P300 latency, a positive peak in the ERP waveform associated with attention and detection, was analyzed from 15 electrode sites (FP_1_, FPz, FP_2_, F_3_, Fz, F_4_, F_7_, F_8_, C_3_, Cz, C_4_, P_3_, Pz, P_4_, and Oz) based on the 10–20 International System [60]. These electrodes utilized Ag–AgCl electrodes and were referenced to link mastoids with the ground electrode positioned at the nasion. Electrode impedance was maintained below 10 kΩ to ensure high-quality data. Recordings were sampled at 256 Hz, and bandpass filtered between 0.16 and 70 Hz. EEG data were segmented into epochs spanning from 200 ms before stimulus presentation to 800 ms after. P300 latency was measured for both standard and target stimuli across all electrode derivations. Trials with a voltage exceeding 70 μV in any non-EOG channel or with incorrect responses were excluded from the analysis. Separate averaging was performed for target and non-target stimuli, with baseline correction applied to the 200 ms pre-stimulus period. The time series and spectrum were carefully monitored before and after artifact rejection and artifact control was applied to ensure the validity of the dependent variable of interest. Consequently, only participants with a minimum of 30 artifact-free trials for both types of stimuli were included in further analysis [61]. It should be mentioned that only correct responses to the oddball stimuli were collected during the reaction time process. The entire recording procedure took place in a soundproof room free from distractions [62,63,64].

### 2.3. Remediation Program

All sixteen (16) children diagnosed with depression participated in a four-month remediation program consisting of 16 sessions, with 2 additional sessions held one and three months later. Sessions occurred once a week, lasting 45 min to 1 h. The CBT program included activities to elucidate the child/adolescent about depression, establish a therapeutic relationship, set the structure and content for the remediation, and agree on the goals of the CBT intervention. The CBT also encompassed exposure techniques, guided discovery, cognitive restructuring techniques, behavioral experiments, self-talk, thought interruption, positive imagery, communication skills, scheduling pleasant events with peers, strengthening problem-solving abilities, and relaxation techniques. Additionally, self-help sessions focused on recognizing and appropriately dealing with depression symptoms and training in reinforcement strategies, including logical rewards for gradually facing depressive situations. The CBT program also included family interventions. Parents of depressive children attended ten (10) sessions once a week, separate from their children, lasting 45 min to 1 h. These sessions aimed to address the parents’ own anxiety and/or depression and included strengthening family communication, informative sessions about depression and expectations, role-playing strategies, and cognitive techniques. Parents were also instructed to form a support group. Eight children (four males and four females) participated solely in the CBT program, while the other eight (four males and four females) received both CBT and psychopharmacological support with SSRIs, as recommended by the child and adolescent psychiatrist who diagnosed them.

### 2.4. Statistical Analysis

One-way ANOVAs compared pre-remediation P300 latency in children and adolescents with depression against the control group and post-remediation P300 latency results in children and adolescents with depression against their peers. Non-parametric statistical analysis compared P300 latency in children and adolescents who participated in the CBT program against those who used CBT with medication. Effect sizes using Cohen’s d [65] were calculated and reported. ANOVAs compared reaction time differences before and after remediation against the control group. Descriptive statistics presented results in the CDI test for children and adolescents with depression before and after remediation in comparison to the control group.

## 3. Results

Pre-remediation, one-way ANOVAs compared the results of children with depression against their typically developing peers. Descriptive statistics analyzed mean scores and standard deviations of P300 latency measured at 15 scalp sites (e.g., Figure 1 and Figure 2), and the same procedure was followed for reaction time measurement. Cohen’s d-effect sizes were calculated. Table 1 presents the mean scores, standard deviations, statistical significance, and effect size of P300 latency from all recorded brain areas.

Table 1 shows that children and adolescents diagnosed with depression had longer latencies in the P300 waveform across all topographic brain regions compared to the control group (*p* < 0.05). The effect sizes ranged from 1.19 to 6.40. Effect sizes below 0.20 are regarded as modest, between 0.20 and 0.50 as medium, and larger than 0.50 as large in magnitude [65]. The reaction time was recorded as participants with depression and their average peers had to push a button in a joystick when the oddball stimuli were presented. The reaction time for children with depression was 429.75 ms (SD 43.06 ms), and their average peers responded at 334.54 ms (SD 23.55 ms). The ANOVA revealed that children with depression presented larger latencies in reaction time compared to children who participated in the control group (*p* < 0.01). In the CDI test, all children and adolescents with depression recorded scores ≥ 22 compared to their typically achieving peers who scored ≤13.

Post remediation, one-way ANOVAs were performed to assess the outcomes of children with depression compared to their typically developing counterparts. Descriptive statistics were utilized to examine the average scores and deviations from the mean of P300 latency measured across 15 scalp locations (e.g., Figure 3 and Figure 4), with a similar approach taken for reaction time measurement. Cohen’s d-effect sizes were computed. Table 2 displays the average scores, standard deviations, statistical significance, and effect size of P300 latency across all recorded brain regions.

As Table 2 indicates, no statistically significant differences were detected in the P300 latency in these post-remediation comparisons. The evidence suggests that the remediation program (CBT and combination of CBT with SSRIs) proved successful in improving the cognitive profile of children and adolescents with depression, as P300 latency between the two groups of children shows no statistically significant difference *p* > 0.05). A careful study of the calculated effect sizes reveals that in 12 out of 15 P300 waveforms, the differences between the two groups were small to medium, ranging from 0.02 to 0.40. It must be mentioned that only in the Fp2, F8, and P4 brain regions did the effect size reach a large magnitude, ranging from 0.48 to 0.54. 

The reaction time for children with depression was recorded at 325.09 ms (SD 35.78 ms), while the control group had a reaction time of 322.01 ms (SD 28.22 ms) to the oddball stimuli. The ANOVA indicated no significant difference (*p* > 0.05) in reaction times between the depressed children and the control group. In the CDI test, children and adolescents with depression had a mean score of less than or equal to 15, compared to less than or equal to 13 for their typically developing peers.

The analysis then specifically examined differences in P300 latencies among children and adolescents with depression based on their participation in the remediation program, which included either only CBT or a combination of CBT and SSRIs. A Mann–Whitney non-parametric statistical analysis was performed, with eight children participating in the CBT program alone and eight receiving both CBT and SSRIs. The results are shown in Table 3, which presents the mean scores, standard deviations, and statistical significance of the P300 latency for participants with depression based on the remediation program they followed. Because a non-parametric method of statistical analysis was used, the effect size was not computed.

As shown in Table 3, children and adolescents with depression who participated in the CBT-only program exhibited longer P300 latencies compared to those who participated in the combined CBT and medication program. However, the analysis found no significant difference (*p* > 0.05). Additionally, a one-way ANOVA was conducted to compare the reaction times between the two groups. Children and adolescents in the combined CBT and medication program had a reaction time of 330.72 ms (SD 20.48 ms), while those in the CBT-only program had a reaction time of 338.84 ms (SD 22.81 ms). These results were also not statistically significant (*p* > 0.05). In summary, the overall results suggest positive outcomes for both remediation programs. The implications of this study are discussed in the next section.

## 4. Discussion

This study aimed to evaluate the electrophysiological brain activity of children and adolescents diagnosed with depression for the first time by examining the P300 waveform latency in Event-Related Potentials (ERPs), reaction time, and CDI scores before and after a remediation program. Additionally, their results were compared to those of their typically developing peers.

The first hypothesis, grounded in previous research [25,26,27,28,29,30,31,32,33,34,35,36,37], posited that children and adolescents with depression would exhibit longer P300 latencies and reaction times compared to a control group. This study confirmed this hypothesis, revealing statistically significant (*p* < 0.05) prolonged P300 latencies in individuals with depression. These findings align with prior studies, suggesting that the extended P300 latency observed in depressed children and adolescents reflects underlying cognitive impairments [66].

Studies reveal that P300 components reflect the influence of the noradrenergic system, and there is evidence suggesting a connection to the dopaminergic system [67,68]. P300 components are thought to represent neural activity originating from the locus coeruleus–noradrenaline system, particularly in response to stimuli with motivational significance. This hypothesis posits that increased activation of the locus coeruleus, resulting in noradrenaline release in cortical regions, is manifested in the P300 waveform [69]. Given the associations between the P300 response and neurotransmitter systems pertinent to conditions like depression and functions such as motivation and attention, the P300 component holds promise as a biomarker for depression [34,70].

Marchetti et al. [71] conducted a meta-analytic study that found a robust association between depression and memory, highlighting a preferential recall of depressive information in individuals at risk of developing major depression. Cognitive deficits in attention, memory, and executive function are frequently reported in depressed individuals [35,40]. In this study, effect sizes ranging from 1.19 to 6.40 indicated substantial differences in P300 latencies between the depressed and control groups [65]. Additionally, reaction times were significantly longer in depressed participants, consistent with other studies [68,69], likely due to diminished attention and interest in tasks, reflecting their clinical presentation [60]. Higher scores on the Children’s Depression Inventory (CDI) among depressed children further supported the first hypothesis.

The second hypothesis was based on existing research on brain plasticity and neural cortex activation, which shows variations in P300 latencies and reaction times following the administration of Cognitive Behavioral Therapy (CBT) and/or selective serotonin reuptake inhibitors (SSRIs). In this study, children and adolescents with depression demonstrated shorter P300 latencies and reaction times after undergoing the remediation program. Post-treatment, no significant differences in P300 latency and reaction time (*p* > 0.05) were observed compared to their typically developing peers. Effect sizes revealed small to medium differences (0.02 to 0.40) across 12 of 15 brain sites, indicating minor variations between the groups. The improvements in P300 latency and reaction time suggest enhanced memory and attentional functions, with participants responding more confidently post-remediation [71,72,73,74]. Numerous studies (e.g., [43,75,76,77,78]) corroborate the significant clinical benefits of CBT and family CBT programs in treating depression in both adults and children.

The third hypothesis examined the differences in P300 latencies and reaction times based on the type of remediation program—either CBT alone or CBT combined with SSRIs. Analysis revealed that children who underwent the combined treatment exhibited better P300 latency and reaction times compared to those who received only CBT. This suggests that neurotransmitters like serotonin play a crucial role in both depression and the P300 waveform. However, no statistically significant differences (*p* > 0.05) were found, partially supporting the third hypothesis.

Selective serotonin reuptake inhibitors (SSRIs) are a well-established, evidence-based intervention that significantly improves symptoms and functioning [79,80]. They are safe, well-tolerated in children and adolescents, and widely accessible. When combined with CBT, SSRIs consistently yield the most positive outcomes for children and adolescents with depression [81]. The American Psychiatric Association’s practice guidelines for treating Major Depressive Disorder recommend this combined approach [49].

It is worth mentioning that, based on the findings of a critical review and meta-analysis [78], CBT that includes behavioral activation and cognitive restructuring components, like the remediation program implemented in the present study, may improve long-term outcomes for children. Additionally, in the current research, parents were included, as it is suggested [78] that involving caregivers in CBT treatment for children and adolescents may enhance long-term improvement in depression symptoms for various reasons. First, the quality of the attachment relationship is an etiological factor in the development of depression, and children learn adaptive coping strategies. Caregiver involvement also demonstrates a commitment to the intervention. Second, caregiver involvement can facilitate the transfer of control, where caregivers pass on information and skills to the child, potentially increasing intervention fidelity and making the child more aware of the therapeutic process between sessions [78].

This study focused exclusively on the P300 waveform, commonly used for assessing higher mental abilities. The small sample size may have limited the statistical power of the analysis. Despite this, this study’s strength lies in the numerous statistically significant results obtained. Efforts were made to include drug-free children and adolescents who had not undergone any psychotherapeutic protocol, as all participants were receiving their first diagnosis of depression. Furthermore, the study results are consistent with those of previous research, regardless of the sample size. This alignment is observed in studies utilizing large samples (e.g., [26]) as well as those with small sample sizes (e.g., [25,55,81]). This consistency across varying sample sizes indicates that the observed effects in the current study are unlikely to be artifacts resulting from the small sample size (*n* = 16). Another limitation of this study can be the division of the sample into two groups, each consisting of eight children. One group participated solely in the CBT program, while the other group received both the CBT program and psychopharmacological support. This variation in treatment introduces potential confounding factors that could impact the study’s findings. Despite this variation in treatment, the combination of CBT and SSRIs in one group did not produce different outcomes compared to CBT alone. It should be noted that the results of the present study are consistent with those of previous research using analogous treatment methodologies (e.g., [52]). This consistency with existing literature supports the observed effects. 

Behaviorally, this study discovered that children and adolescents with depression exhibited prolonged P300 latencies compared to the control group, suggesting deficits in attention [82] and memory [24] that might account for their subpar academic performance. Research indicates that increased P300 latency is linked to attention and memory impairments. The observed reduction in P300 latency following remediation points to enhancements in higher cognitive functions. Additionally, the decrease in reaction times after the remediation program may indicate positive changes in cognitive behavior and better organization [83]. These findings can contribute to the development of intervention programs in educational settings.

## 5. Conclusions

Despite this study’s limitations, the findings provide valuable insights for assessing children with depression and contribute to the scientific community’s efforts to develop and implement better therapeutic programs for children and adolescents with this diagnosis. This study underscores the significance of electrophysiological measures and cognitive assessments in understanding and remediating depression in children and adolescents. While the current research provides valuable insights, further studies with larger sample sizes, additional ERP components, and longitudinal follow-ups are essential to enhance our understanding of depression and improve therapeutic interventions.

## Figures and Tables

**Figure 1 brainsci-14-00660-f001:**
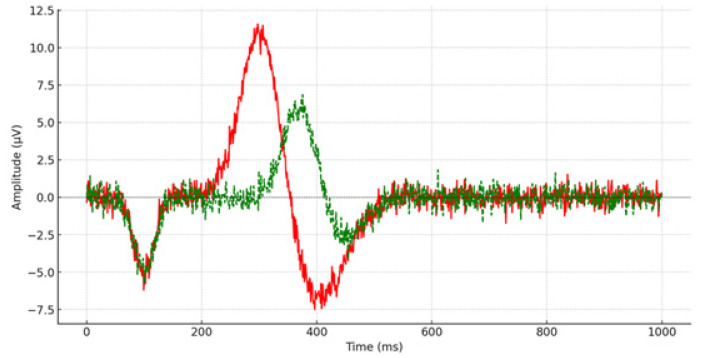
P300 latency from the left prefrontal lobe (FP_1_) during the pre-remediation procedure. The red line depicts the P300 waveform for the control group, while the green line illustrates the P300 waveform for children and adolescents diagnosed with depression.

**Figure 2 brainsci-14-00660-f002:**
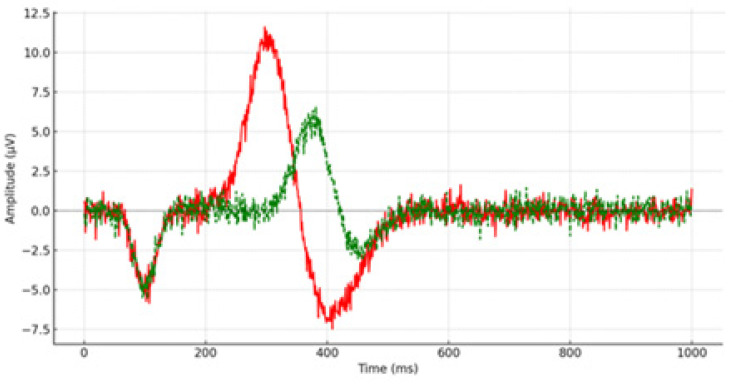
P300 latency from the right prefrontal lobe (FP_2_) during the pre-remediation procedure. The red line depicts the P300 waveform for the control group, while the green line illustrates the P300 waveform for children and adolescents diagnosed with depression.

**Figure 3 brainsci-14-00660-f003:**
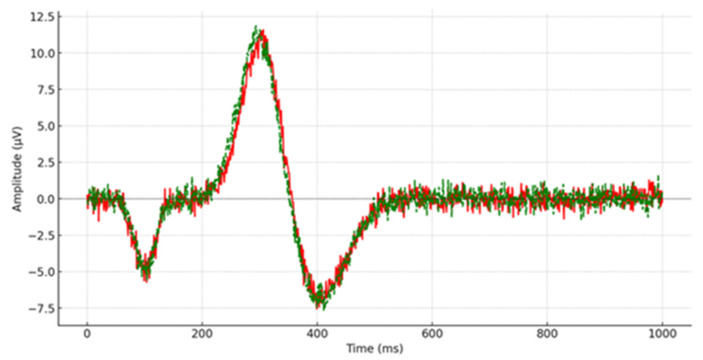
P300 latency from the left prefrontal lobe (FP_1_) during the post-remediation procedure. The red line depicts the P300 waveform for the control group, while the green line illustrates the P300 waveform for children and adolescents diagnosed with depression.

**Figure 4 brainsci-14-00660-f004:**
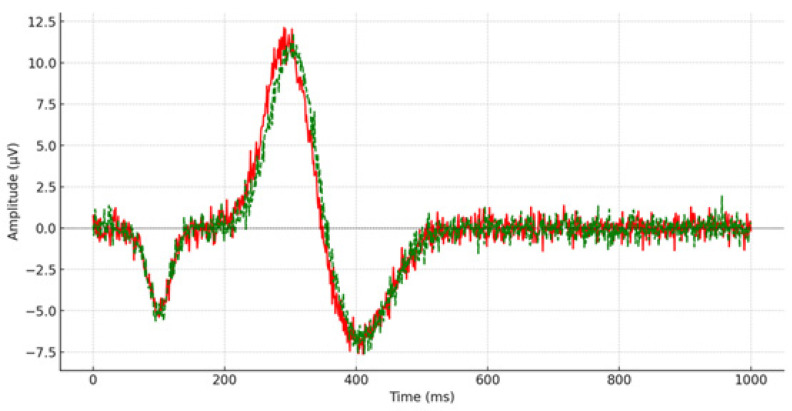
P300 latency from the right prefrontal lobe (FP_2_) during the post-remediation procedure. The red line depicts the P300 waveform for the control group, while the green line illustrates the P300 waveform for children and adolescents diagnosed with depression.

**Table 1 brainsci-14-00660-t001:** Mean scores, standard deviations, significance, and effect size of P300 latency for children and adolescents who participated in the control group and children and adolescents with depression pre-remediation.

Electro/ Encephalographic Sites	P300 Latency of Control Group		P300 Latency of Participants with Depression Pre-Remediation				
	M	SD	M	SD	F	Sign.	Cohen’s d
FP_1_	304.28	13.74	377.11	13.64	64.247	0.001	5.32
FP_Z_	306.67	13.72	381.20	16.57	58.237	0.001	4.89
FP_2_	313.02	8.22	373.95	10.66	58.936	0.001	6.40
F_3_	310.41	10.64	365.28	9.98	75.102	0.001	5.32
F_Z_	316.24	11.12	372.01	19.41	99.454	0.001	3.55
F_4_	320.20	10.31	363.96	14.10	100.391	0.001	3.54
F_7_	311.97	10.47	375.57	15.31	86.294	0.001	4.85
F_8_	318.99	9.03	364.65	14.71	48.201	0.001	3.74
C_3_	319.50	12.28	351.15	15.23	41.882	0.001	2.29
C_Z_	325.58	7.38	346.56	16.01	22.684	0.001	1.68
C_4_	328.36	14.25	347.65	17.99	11.316	0.002	1.19
P_3_	314.30	15.34	352.15	18.68	21.659	0.001	2.21
P_Z_	321.88	12.87	349.90	11.36	20.211	0.001	2.31
P_4_	326.06	8.72	346.48	17.67	17.283	0.001	1.46
O_Z_	321.36	9.05	336.56	14.44	5.436	0.027	1.26

**Table 2 brainsci-14-00660-t002:** Mean scores, standard deviations, significance, and effect size of P300 latency for children and adolescents who participated in the control group and children and adolescents with depression post-remediation.

Electro/ Encephalographic Sites	P300 Latency of Control Group		P300 Latency of Participants with Depression Post-Remediation				
	M	SD	M	SD	F	Sign	Cohen’s d
F_P1_	297.47	9.83	295.90	5.38	1.523	0.235	0.20
F_PZ_	303.14	8.95	300.43	6.62	1.113	0.342	0.34
F_P2_	307.21	8.22	303.26	6.33	1.417	0.259	0.54
F_3_	304.21	9.99	308.12	9.51	0.538	0.590	0.40
F_Z_	307.77	9.66	311.26	7.94	0.424	0.658	0.39
F_4_	312.76	9.24	313.64	9.12	0.044	0.957	0.09
F7	306.72	11.47	303.80	8.09	1.029	0.370	0.29
F8	314.71	11.65	309.78	8.63	1.942	0.162	0.48
C3	319.41	6.50	317.02	7.31	0.654	0.527	0.34
CZ	322.58	5.93	324.21	6.82	0.262	0.771	0.25
C_4_	326.28	6.58	326.11	7.53	0.014	0.986	0.02
P_3_	324.47	6.38	321.61	7.79	2.202	0.129	0.40
P_Z_	323.18	7.08	323.46	6.17	1.303	0.287	0.04
P_4_	329.86	7.02	326.28	8.47	1.161	0.327	0.46
O_Z_	323.48	6.19	325.94	12.82	0.584	0.564	0.24

**Table 3 brainsci-14-00660-t003:** Mean scores, standard deviations, and significance of P300 latency between children and adolescents with depression who underwent CBT and SSRIs and children and adolescents who underwent only CBT in the remediation program.

Electro/ Encephalographic Sites	P300 Latency of Children and Adolescents That Followed CBT and Medication		P300 Latency of Children and Adolescents That Followed only CBT Program		
	M	SD	M	SD	U
F_P1_	291.36	3.07	295.90	5.37	0.071
F_PZ_	297.95	6.62	300.43	6.62	0.758
F_P2_	301.65	7.55	303.25	6.33	1.000
F_3_	308.12	9.51	308.26	14.01	1.000
F_Z_	310.40	11.85	311.26	7.94	1.000
F_4_	313.64	9.12	314.04	11.05	1.000
F_7_	300.10	9.86	303.81	8.09	0.351
F_8_	305.69	9.63	309.78	8.63	0.351
C_3_	317.02	4.31	318.13	4.83	0.299
C_Z_	322.28	4.95	324.21	4.42	0.408
C_4_	326.09	4.09	326.11	4.72	1.000
P_3_	319.55	4.79	321.61	3.67	0.470
P_Z_	320.52	4.17	323.46	3.81	0.114
P_4_	325.64	7.07	326.28	7.44	1.000
O_Z_	321.29	5.19	323.50	4.34	0.299

## Data Availability

The original contributions presented in the study are included in the article, further inquiries can be directed to the corresponding author.

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
