# Peer review of "Differences in Children and Adolescents with Depression before and after a Remediation Program: An Event-Related Potential Study"

_brainsci, 2024, doi:10.3390/brainsci14070660_

Round 1

Reviewer 1 Report

Comments and Suggestions for Authors

This paper studies the effectiveness of Cognitive Behavioral Therapy (CBT) alone and CBT combined with selective serotonin reuptake inhibitors (SSRIs) in a remediation program of depression treatment as an example of major depressive disorder (MDD). The studied sample included 16 subjects with depression symptoms in two groups versus 16 healthy volunteers with similar demographic characteristics; the methodology used Event-Related Potentials (ERPs), the Children Depression Inventory (CDI), and reaction time tests. Participants with participants with depression had cognitive deficits in attention and memory, indicated by longer P300 latencies. The authors are 

 convinced about the effectiveness of CBT alone or CBT combined with medication; the children and adolescents with depression exhibited lower CDI scores and shorter P300 latencies and reaction times compared to test results before therapy and the test parameters relative to the control group.

The presented State-of–art in the Introduction section convinces about the necessity of undertaking the project, the appropriateness of the applied methodology in the study (especially of the clinical neurophysiology); the selection of the literature on the problem is reasonable. Previous studies have found executive function, memory, psychomotor speed, and attention problems in MDD patients worse compared to healthy controls. The aims/hypotheses are clearly formulated; the significance of the P300 abnormities detection as the MDD depression marker sounds.

The authors concluded about the necessity to develop and implement better therapeutic programs for children and adolescents based on the P300 evaluation, which is true considering the high confidence of the measurable parameter.

Minor corrections are required:

1. One of the study limitations is the small sample size, which may have lowered statistical analysis power, therefore the additional Bonferroni correction application can be advised for the authors. They admitted this limitation in the Discussion section.

2. More data on the symptoms and incidence of depression is required in the groups of patients for their better characterisation.

3. To present the study’s results more clearly for the neurologists and the neurophysiologists, examples of representative P300 recordings in three groups of subjects for comparison should be presented in a Figure. Data in a Table are a little convincing.

4. Conclusion subchapter can be introduced in lines 390-398

5. Refs. require the order citation typical for MDPI; some include the typographic mistakes e,g. 2, 3, etc. as well as in the text, e.g. line 355.

Comments on the Quality of English Language

Minor corrections are required

Author Response

Dear reviewer,

Thank you for your thorough evaluation of our study and your insightful comments, which have significantly enhanced the quality of our work. We provide our responses in red text.

Reviewer 2 Report

Comments and Suggestions for Authors

The article focuses on investigating the effectiveness of Cognitive Behavioral Therapy (CBT) alone and CBT combined with selective serotonin reuptake inhibitors (SSRIs) in a remediation program. The author states that the main aim of the research was to compare the electrophysiological brain responses of children and adolescents diagnosed with depression for the first time at a state child and adolescent psychiatric hospital, with those of their typical peers using Event-Related Potentials (ERPs). I advise the author to frame the aim more as a problem statement and avoid being overly technical.

The topic addressed in this article is both important and relevant from fundamental and applied perspectives. The author provides a detailed and high-quality literature review. However, a specific research gap is not explicitly identified. I recommend the author strengthen this aspect in the article.

Compared to other published material, this study contributes new information about the dynamics of P300 latency before and after the intervention.

I have no questions regarding the research methodology.

In the Discussion section, the author should enhance the comparison of the obtained results with previously published studies. Currently, there are only a few references to literary sources in the Discussion section.

The references are appropriate.

Reviewer 3 Report

Comments and Suggestions for Authors

“Differences in children and adolescents with depression before and after a remediation program: An Event Related Potentials Study”( brainsci-3050756

This manuscript aimed to evaluate the effectiveness of Cognitive Behavioral Therapy (CBT) alone and CBT combined with selective serotonin reuptake inhibitors (SSRIs) in a remediation program on P300 in children and adolescent. The results revealed that participants with depression had cognitive deficits in attention and memory, indicated by longer P300 latencies at the baseline. After undergoing remediation with either CBT alone or CBT combined with medication, the children and adolescents with depression exhibited lower CDI scores and shorter P300 latencies and reaction times compared to before remediation and relative to the control group. Overall, this topic is interesting and the results hold some practical implications. However, some concerns appeared after reading the whole manuscript, especially the methodology part, the writing logic and flow, and the theoretical implications.

1. Some important papers need to be reviewed and discussed, such as,

Related reviews about P300 and MDD:

Arıkan, M. K., İlhan, R., Orhan, Ö., Esmeray, M. T., Turan, Ş., Gica, Ş., … Metin, B. (2024). P300 parameters in major depressive disorder: A systematic review and meta-analysis. The World Journal of Biological Psychiatry, 25(4), 255–266. https://doi.org/10.1080/15622975.2024.2321554

Kangas, E. S., Vuoriainen, E., Lindeman, S., & Astikainen, P. (2022). Auditory event-related potentials in separating patients with depressive disorders and non-depressed controls: a narrative review. International Journal of Psychophysiology, 179, 119-142.

Directly-related empirical studies

Dickey, L., Pegg, S., Cárdenas, E.F. et al. Neural Predictors of Improvement With Cognitive Behavioral Therapy for Adolescents With Depression: An Examination of Reward Responsiveness and Emotion Regulation. Res Child Adolesc Psychopathol 51, 1069–1082 (2023). https://doi.org/10.1007/s10802-023-01054-z

Burkhouse, K. L., Gorka, S. M., Klumpp, H., Kennedy, A. E., Karich, S., Francis, J., ... & Phan, K. L. (2018). Neural responsiveness to reward as an index of depressive symptom change following cognitive-behavioral therapy and SSRI treatment. The Journal of clinical psychiatry, 79(4), 12587.

Zhou, L., Wang, G., & Wang, H. (2018). Abnormalities of P300 before and after antidepressant treatment in depression: an ERP-sLORETA study. Neuroreport, 29(3), 160-168.

Linka, T., Sartory, G., Wiltfang, J., & Müller, B. W. (2009). Treatment effects of serotonergic and noradrenergic antidepressants on the intensity dependence of auditory ERP components in major depression. Neuroscience letters, 463(1), 26-30.

2. The writing logic and flow is rather confusing and hard to follow. For example,

The fourth paragraph about brain structure and function only related not so much with the rest of the formal context of the manuscript.

Line 109-115 does not make sense to the manuscript and provide limited information about the current topic.

The introduction part needs to be focused more on the P300 related findings about MDD.

The sixth paragraph is rather confusing because no previous P300 research findings were mentioned before this paragraph and then the author stated that There have been studies that have used ERPs as a method of assessing depression suggesting inconsistent results. More specifically, a previous study found no significant difference in the latency of the P300 waveform [24].

3. The hypothesis were not well formulated in the introduction part and the number of each hypothesis need to be clearly identified to help the authors to correspond the number of hypothesis in the discussion part to those in the introduction part.

4. The latest prevalence of MDD in children and adolescents needs to be updated.

5. How did you determine the sample size? Did you calculate the sample size needed before formal study?

The current sample size seems too little to get reliable results.

6.3. Statistical Analysis” should be 2.4 Statistical Analysis. In this part, how did you treat outliers of reaction time should be mentioned.

7. It should also be noted that not only the latency but also the amplitude of P300 would be useful to differential adolescent with depression and those without depression, thus, it would also be interesting to see the results of amplitude of P300.

8. Please follow the publication guideline of EEG studies.

Keil, A., Bernat, E. M., Cohen, M. X., Ding, M., Fabiani, M., Gratton, G., ... & Weisz, N. (2022). Recommendations and publication guidelines for studies using frequency domain and timefrequency domain analyses of neural time series. Psychophysiology, 59(5), e14052.

9. The waveforms for ERPs should be provided.

10. P300 latency” should be mentioned in the caption of table 2 and 3.

11. For the p value, please provide the specific data unless it is less than 0.001.

12. The analysis revealed that depressed children that implemented the combined program exhibited better P300 latency and better reaction time in comparison to depressed children that were treated only with a CBT program.please to replace “better” with more informative words.

13. The theoretical implications of the current findings need to be discussed more in the current manuscript.

Hajcak, G., & Foti, D. (2020). Significance?... Significance! Empirical, methodological, and theoretical connections between the late positive potential and P300 as neural responses to stimulus significance: An integrative review. Psychophysiology, 57(7), e13570.

14. “It should be noted that 8 children (4 males and 4 females) were only engaged in the CBT program, whereas another 8 children (4 male and 4 female) participated in both the CBT program and psycho-pharmacological support. The child and adolescent psychiatrist who diagnosed them suggested SSRIs.” This treatment would further confound the findings.

15. I strongly recommend that the paper be thoroughly proofread and edited for languages and grammars, to enhance readership.

Comments on the Quality of English Language

 Extensive editing of English language required

Round 2

Reviewer 2 Report

Comments and Suggestions for Authors

I can see that the authors have revised the paper according to my suggestions.

My only advice is to proofread the text.

Comments on the Quality of English Language

I see some errors in the text. For example:

'The findings are in the same line studies [51-56] using both CBT and SSRIs in the treatment of patients with depression.'

This article needs proofreading.

Reviewer 3 Report

Comments and Suggestions for Authors

Thanks for the revisions which significantly improve the quality of the manuscript, however, some concerns still remain.

1. For the p value, I suggest to add one row for p values in table 1 and 2.

2. For the sample size determination, it seems that you did not calculate the sample size needed before formal study according to your response. Then the small sample size might be a potential limitation which needs to be discussed in the limitation part.

3. For the comment 14, the methodology cannot be justified just because you used this method before. And the confounding effects cannot be excluded. Thus, this point needs to be included as a potential limitation and needs to be discussed in the discussion part as well.

Comments on the Quality of English Language

 Minor editing of English language required
